# The Moderated-Mediation Effect of Workplace Anxiety and Regulatory Focus in the Relationship between Work-Related Identity Discrepancy and Employee Innovation

**DOI:** 10.3390/ijerph17176121

**Published:** 2020-08-23

**Authors:** Chang-E Liu, Chenhong Hu, Wei Xie, Tingting Liu, Wei He

**Affiliations:** 1Mobile E-business Collaborative Innovation Center of Hunan Province, Key Laboratory of Hunan Province for Mobile Business Intelligence, College of Business Administration, Hunan University of Technology and Business, Changsha 410205, China; liuce15@hutb.edu.cn; 2College of Business Administration, Hunan University of Technology and Business, Changsha 410205, China; 201910310082@stu.hutb.edu.cn (C.H.); 70107@hutb.edu.cn (T.L.); 3Scott College of Business, Indiana State University, Terre Haute, IN 47809, USA

**Keywords:** work-related identity discrepancy, employee innovation behavior, workplace anxiety, regulatory focus

## Abstract

Extant research on work-related identity discrepancy mostly has probed its effects on employees’ attitudes and emotions but has paid little attention to its impact on employee behaviors. Drawing on self-discrepancy theory, we examined the influencing mechanism and conditions of work-related identity discrepancy on employee innovation behavior. With data collected from 563 employees who personally experienced leadership transition in the workplace, we found that work-related identity discrepancy predicts employee innovation behavior through workplace anxiety. We also discovered that employees’ personality traits—promotion regulatory focus and prevention regulatory focus in particular—can intensify or buffer the negative relationship between work-related identity discrepancy and employee innovation behavior. We further discuss the conceptual and practical implications of these findings.

## 1. Introduction

In an era of volatility, uncertainty, complexity and ambiguity (VUCA) [1], leadership changes in organizations are more common than ever before [2]. Both organizational and individual performance may fluctuate during leadership transition since employees, as well as managers, have to adapt to changed contexts [3]. Particularly, employees who had a good relationship with their previous leader are prone to work-related identity loss or discrepancy [4,5,6]. As part of an individual’s overall self-discrepancy, work-related identity discrepancy refers to the phenomenon that external events break the cognitive balance of existing work identity and lead to a gap between one’s actual self and the ideal or ought self [6,7]. Extant studies on work-related identity discrepancy have mostly focused on employees’ attitudes (e.g., turnover thought [5], job disengagement [6], productivity loss and emotional labor [8]) and emotions (e.g., anxiety and stress [9]) as consequences of work-related identity discrepancy but have largely ignored its impact on employee behaviors—what employees actually do in their jobs or what actions they take under certain circumstances in the workplace. One such employee behavior is innovation behavior, as it is the foundation of organizational innovation, which in turn plays a crucial role in the survival and development of organizations [10]. Since identity discrepancy is a key source of depression, frustration, shame and anxiety according to self-discrepancy theory [11,12], we believe that work-related identity discrepancy can trigger workplace anxiety and further affects employee innovation behavior. Meanwhile, individuals’ different traits affect the degree of emotion reaction. Previous research found that employees’ regulatory focus shapes how they perceive their environment and how they respond to it. Promotion regulatory focus and prevention regulatory focus, in particular, are likely to influence employees’ emotions (e.g., anxiety, depression, guilt) [13,14]. That is, when facing work-related identity discrepancy, employees with different regulatory focus perceive workplace anxiety differently. Our study explores the mediating role of workplace anxiety as well as the moderating role of regulatory focus in the relationship between work-related identity discrepancy and employee innovation behavior.

*Work-related identity discrepancy and employee innovation behavior*. Previous research has found that individuals who identify with a group can acquire a sense of security and accomplishment and therefore boost their work engagement and innovation behavior [15]. Other research has also pointed out that, when employees feel a loss of work-related identity or a threat to their status in a group, they face such negative outcomes as demotivation, job dissatisfaction, insecurity and disengagement [16,17,18]. Specific to our research context, employees who have a good relationship with their previous leader may feel that they no longer belong to the insider group of the new leader after leadership transition in their workplace. The loss of insider identity leads to the employees’ work-related identity discrepancy [19] and makes them feel tired, both cognitively and emotionally [14], and less willingly to innovate [20]. Previous studies have also revealed that employees who are “insiders” of the previous leader tend to perceive more work-related identity discrepancy after the leader departs [5,6]. They are reluctant to give up their previous privilege as the insiders in terms of work opportunities, resources and support [21], which are helpful to their innovation behavior. Consequently, they are prone to cutting their innovation behaviors. Furthermore, self-discrepancy theory points out that individuals maintain self-guides, defining their ideal selves (beliefs about the attribute “I would like to possess”) and ought selves (beliefs about the attribute “I should possess”) [22]. It also suggested people tend to feel uneasy and stressed when they find a discrepancy between their actual self and their ideal self or ought self; they will take actions to narrow the gap [22]. They are likely to dedicate plenty of time and energy to catering to their new leader and building a close relationship with him or her. As a result, however, they are unable to devote adequate efforts to work innovation, or they may lower the standards of the ideal self or ought self, leading to unwilling attitudes in innovation behavior. Based on the discussion, we predict:

**Hypothesis** **1 (H1).**
*Work-related identity discrepancy*
*has a negative effect on employee innovation*
*behavior.*


*Mediating effect of workplace anxiety.* Workplace anxiety is an employee’s feeling of nervousness, uneasiness and tension about job-related factors [23,24]. Official statistics have indicated that 55% of Americans report feeling anxious during the workday [25]. These data raise serious concerns, as workplace anxiety is closely associated with a series of negative consequences, such as unethical behaviors [26], poor job performance [23] and risk-taking behaviors [27]. Moreover, a meta-analytic review of 59 independent samples found that anxiety is significantly and negatively related to creative performance [28]. Eysenck and colleagues suggested that workplace anxiety may hinder employee innovation behavior through influencing their cognition—more anxious individuals think less efficiently than calmer people [29]. However, any workplace has features that provoke anxiety [30]. A prior leader’s departure can make employees worried about the possible loss of vested interests like resources and opportunities [31]. Moreover, the new leader brings in various uncertainties [32] in terms of leadership style and new insider membership. All those worries and uncertainties from work-related identity discrepancy may cause employee workplace anxiety. Additionally, self-discrepancy theory demonstrates that a self-discrepancy between the actual self and the ideal or ought self can cause negative emotions like anxiety, guilt and depression in the individual, which then affect people’s behaviors [33,34]. Work-related identity discrepancy may prevent employees from doing their best job as they did in the old leader’s team (the ideal or ought self), thus contributing to negative emotions (e.g., workplace anxiety) that in turn negatively affect innovation behavior. Summarizing all the discussion so far, we believe that employees perceiving work-related identity discrepancy will experience workplace anxiety and engage in less innovation behaviors. Therefore, we propose:

**Hypothesis** **2 (H2).**
*Workplace anxiety mediates the effect of work-related identity discrepancy on employee innovation behavior.*


*Moderating effect of regulatory focus.* Individuals achieve their goals through two self-regulatory motivational systems—promotion regulatory focus and prevention regulatory focus [34]. Promotion regulatory focus is a motivational condition that is sensitive to the presence or absence of positive outcomes, while prevention regulatory focus emphasizes the need for security and the fulfillment of duties and obligations through vigilant and responsible behaviors [13,35]. Different regulatory foci have different sensitivities to and experience diverse emotions towards certain information [36], such as work-related identity discrepancy. Employees with promotion regulatory focus are typically high achievers and more sensitive to the presence or absence of positive outcomes [13,34]. To them, work-related identity discrepancy means that they are no longer the insiders of the new leader. Thus, their resources and opportunities could markedly drop after leadership transition [5,6]. They are more likely to experience workplace anxiety since losses of resources or opportunities become barriers to their work performance and career success. In contrast, people with prevention regulatory focus are typically conservative. They care more about taking responsibility, fulfilling their obligations, preventing loss and avoiding mistakes [13,35]. Employees with higher prevention regulatory focus tend to take a cautious approach to regulating their behavior instead of pursuing higher goals; they can quickly adapt to environmental changes and keep doing their job as well as before [13]. To them, work-related identity discrepancy and status threat in the organization will not make them feel more anxious since they are not keen on work achievement. Meanwhile, self-discrepancy theory demonstrates that identity discrepancy will trigger emotional reactions [14], and different traits (such as regulatory focus) will generate different degrees of emotional response [37]. All in all, we believe that employee regulatory focus may change the effect of work-related identity discrepancy on workplace anxiety. Thus, we propose a pair of moderation hypotheses:

**Hypothesis** **3a (H3a).**
*Employees with promotion regulatory focus moderate the positive effect of work-related identity discrepancy on workplace anxiety, such that the positive effect is stronger for employees with high levels of promotion regulatory focus.*


**Hypothesis** **3b (H3b).**
*Employees with prevention regulatory focus moderate the positive effect of work-related identity discrepancy on workplace anxiety, such that the positive effect is weaker for employees with high levels of prevention regulatory focus.*


*Moderated mediation model.* Based on the hypotheses above, our research further develops a moderated mediation model. That is, promotion regulatory focus and prevention regulatory focus moderate the mediating effect of workplace anxiety on the relationship between work-related identity discrepancy and employee innovation behavior, respectively. We further propose another pair of hypotheses:

**Hypothesis** **4a (H4a).**
*The indirect effect of work-related identity discrepancy on employee innovation behavior via workplace anxiety is moderated by promotion regulatory focus, such that the indirect effect will be strengthened for employees with high levels of promotion regulatory focus.*


**Hypothesis** **4b (H4b).**
*The indirect effect of work-related identity discrepancy on employee innovation behavior via workplace anxiety is moderated by prevention regulatory focus, such that the indirect effect will be weakened for employees with high levels of prevention regulatory focus.*


Altogether, we summarize our research variables and hypotheses in a conceptual framework in Figure 1.

## 2. Materials and Methods

*Sample and design.* We had our research proposal approved by the Academic Ethic Committees of our institutions (IRB 862-898) before collecting data. To test our hypotheses, we adopted a cross-sectional research design [38,39] and surveyed employees who personally experienced direct leadership transition (or change of boss) in their workplace. Data were collected through onsite and online surveys. Onsite surveys were conducted in the part-time MBA programs of two universities located in a metropolis in southern China. These MBA students were full-time employees of organizations in the manufacturing, tech, services, government and non-profit sectors. We collected a total of 237 valid responses out of 300 participants (response rate 79.00%) onsite. Following the example of Zhou, Deng and Rao [40], we also provided our survey online through SoJump.com, a popular professional online survey platform. We received 326 valid responses online out of 500 invited participants (response rate 65.20%), who came from various organizations similar to the onsite survey. All participants completed the survey on a voluntary basis and were assured of the anonymity and confidentiality of their responses. Finally, a total of 563 valid responses were received out of 800 participants (response rate 70.37%), including 287 men (50.98%) and 276 women (49.02%). Among the 563 respondents, 73.71% were under 40 years old (average 33.49, SD = 8.56), 50.62% were average employees, 58.61% had a bachelor’s degree or above and 91.65% had worked longer than one year.

*Measures.* To ensure the reliability and validity of measurements, we adopted well-established scales developed and used by previous researchers. All scale items underwent a back-translation process [41] and used a Likert five-point scale with 5 for “strongly agree” and 1 for “strongly disagree,” except demographic items. We adapted a three-item scale developed by Khan, Moss, Quratulain and Hameed [42] to measure *work-related identity discrepancy*. A sample item is “After experiencing replacement of leadership, I felt I had less control over resources than I had before” (*α* = 0.865). *Workplace anxiety* was evaluated using an eight-item scale developed by Mccarthy, Trougakos and Cheng [23]. A sample item is “I’m afraid I can’t get a good evaluation on job performance” (*α* = 0.928). We measured *employee innovation behavior* with a six-item scale developed by Scott and Bruce [43]. A sample item is “I often come up with creative ideas” (*α* = 0.891). For the two types of regulatory focus, we adopted the six-item scale for *promotion regulatory focus* and the 5-item scale for *prevention regulatory focus* scale, respectively, both of which were developed by Higgins, Friedman, Harlow, Idson, Ayduk and Taylor [44]. A sample item for promotion regulatory focus is “I often do things that motivate me to work harder”) (*α* = 0.911) and an item for prevention regulatory focus is “During the process of my growth, I seldom did things my parents wouldn’t tolerate”) (α = 0.889). We used respondents’ gender, education, age and tenure as control variables so as to conduct a rigorous test of our hypotheses and rule out alternative explanations [45].

*Data analyses.* We used SPSS 22.0 (IBM Corp, Armonk, NY, USA), PROCESS macro (The Ohio State University, Columbus, OH, USA), and MPLUS 7.0 (Muthen & Muthen, Los Angeles, CA, USA) to analyze the data and test our hypotheses.

## 3. Results

*Preliminary analyses.*Table 1 demonstrates the correlation coefficients, means and standard deviations of all variables. As shown in Table 1, work-related identity discrepancy and workplace anxiety have a significantly positive correlation (*γ* = 0.493, *p* < 0.01), work-related identity discrepancy and employee innovation behavior have a significantly negative correlation (*γ* = −0.501, *p* < 0.01) and workplace anxiety and employee innovation behavior have a significantly negative correlation (*γ* = −0.577, *p* < 0.01). These results provide preliminary support for our hypotheses of the main effect and the mediation effect.

We used two established approaches to reduce the influence of common method bias in our study. The first approach was program control, which emphasizes the anonymity and confidentiality of the responses and uses filler items and different instructions to create a psychological separation between the sets of variables [46]. The other approach was statistical control, which conducts a varimax rotation analysis of principal factors for all variables to examine the common method variance. Our data show that the first factor explains 33.62% of the variance, which is less than the recommended explained criterion of 50% [46]. Therefore, common method variance is not a serious issue in our study.

We then conducted a series of confirmatory factor analysis (CFA) to ensure that the five latent variables (work-related identity discrepancy, workplace anxiety, promotion regulatory focus, prevention regulatory focus and employee innovation behavior) have satisfactory discriminant validity. The CFA results indicate that the five-factor model had a good fit to the data, *x*^2^*/df* = 1.204, *RMSEA* = 0.019, *IFI* = 0.993, *CFI* = 0.993, *TLI* = 0.992. *Chi-squared* tests show that the five-factor model is superior to a four-factor model where (a) items for work-related identity discrepancy and workplace anxiety were set to load on one factor, Δ*x*^2^
_[4]_ = 618.57, *p* < 0.01 and (b) items for work-related identity discrepancy and employee innovation behavior were set to load on one factor, Δ*x*^2^
_[4]_ = 568.36, *p* < 0.01. These results provide construct validity evidence for the five latent variables in our research.

*Main effect and mediation effect tests.* We tested the main effect of our research with SPSS 22.0 and the mediation effect with PROCESS macro with 5000 bootstrap samples and a confidence interval (CI) of 95%, as recommended by Preacher and Hayes [47]. After considering the effect of control variables, the main effect of work-related identity discrepancy on employee innovation behavior is significantly negative (*β* = −0.388), with a 95% confidence interval (CI) of (−0.440, −0.336) not including 0 (not shown in Table 2). The mediation effect of workplace anxiety is also significantly negative (*β* = −0.163), with a 95% confidence interval (CI) of (−0.202, −0.127) not including 0 (not shown in Table 2). These results support both Hypotheses 1 and 2.

*Moderation effect test.* We conducted a series of regression analyses to test Hypotheses 3a and 3b. To distinguish the impacts of promotion regulatory focus and prevention regulatory focus, we tested their moderation effects separately. To reduce potential collinearity between work-related identity discrepancy and regulatory focus, all explanatory variables (except demographic variables) were decentralized [48].

As shown in Table 2, Model 4 presents that the interaction between work-related identity discrepancy and promotion regulatory focus is significantly related to workplace anxiety (*β* = 0.212, *p* < 0.001). The R^2^ changes from 0.272 (in Model 2) to 0.315 (in Model 4), thus ΔR^2^ = 0.044, *p* < 0.001. These results suggest that promotion regulatory focus plays a positive moderating role between work-related identity discrepancy and workplace anxiety—when promotion regulatory focus is higher, the positive correlation between work-related identity discrepancy and workplace anxiety is stronger. Hence, Hypothesis 3a is confirmed. By the same token, the interaction between work-related identity discrepancy and prevention regulatory focus is significantly negative for workplace anxiety (*β* = −0.180, *p* < 0.001). The R^2^ changes from 0.292 (in Model 3) to 0.315 (in Model 5), thus ΔR^2^ = 0.032, *p* < 0.001. These results support Hypothesis 3b. That is, the higher the prevention regulatory focus, the weaker the positive correlation between work-related identity discrepancy and workplace anxiety.

Following the procedure suggested by Preacher, Curran and Bauer [49], we drew two schematic diagrams to make the moderation effects of regulatory focus look more intuitive and specific. As shown in Figure 2, when promotion regulatory focus is high, the positive relationship between work-related identity discrepancy and workplace anxiety is stronger (*simple slope* = 0.615, *p* < 0.001). On the contrary, the positive relationship is weaker (*simple slope* = 0.235, *p* < 0.001). As shown in Figure 3, when prevention regulatory focus is high, work-related identity discrepancy has a weaker effect on workplace anxiety (*simple slope* = 0.262, *p* < 0.001), and vice versa (*simple slope* = 0.604, *p* < 0.001).

*Moderated mediation effect test.* We further bootstrapped a confidence interval (CI) of 95% to assess the conditioning effect of the two types of regulatory focus on the relationship between work-related identity discrepancy and employee innovation behavior via workplace anxiety. As shown in Table 3, the indirect effect of work-related identity discrepancy on employee innovation behavior via workplace anxiety is *−*0.052, CI (−0.094, −0.014) and −0.209, CI (−0.261, −0.162), respectively, not including 0, when employees possess high promotion regulatory focus and prevention regulatory focus. In contrast, the indirect effect is −0.198, CI (−0.248, −0.151) and −0.077, CI (−0.121, −0.038), respectively, not including 0. All these results indicate that promotion regulatory focus and prevention regulatory focus moderate the indirect effect. Meanwhile, Hayes pointed out that, since the indirect effects are always significant no matter whether the moderator is high or low, the index criterion must be applied to determine whether the moderated mediation effect is significant [50]. Accordingly, the indices of moderated mediation are found to be significant. Specifically, indices of promotion regulatory focus and prevention regulatory focus are *−*0.073, SE = 0.014, CI (−0.101, −0.046) and 0.066, SE = 0.014, CI (0.041, 0.095), respectively, not including 0. Thus, Hypotheses 4a and 4b are verified.

## 4. Discussion and Implications

In this research, we studied the effect of work-related identity discrepancy on employee innovation behavior, especially the moderated mediation effect of workplace anxiety and two types of regulatory focus. We found that work-related identity discrepancy has a negative impact on employee innovation behavior; workplace anxiety mediates the relationship between work-related identity discrepancy and employee innovation behavior; promotion regulatory focus plays a positive moderation role between work-related identity discrepancy and workplace anxiety, while prevention regulatory focus plays a negative moderation role between work-related identity discrepancy and workplace anxiety; and promotion regulatory focus and prevention regulatory focus respectively enhance and undermine the indirect relationship between work-related identity discrepancy and employee innovation behavior via workplace anxiety.

These findings provide important implications to the existing literature on identity discrepancy research. Prior work has mostly focused on the effects of work-related identity discrepancy on employees’ attitudes and emotions, such as turnover thought [5], job disengagement [6], productivity loss, emotional labor [8] and anxiety and stress [9], while largely ignoring its impact on employee behavior. Our study examined the effect of identity discrepancy on employee innovation behavior and its psychological mechanism in the specific context of a workplace where the old leader with a close relationship to employees departs. Our findings not only broaden the existing literature on outcomes of work-related identity discrepancy but also present a new perspective for future research on identity discrepancy.

Our study also contributes to the growing knowledge body of how work-related identity discrepancy hinders employee innovation behavior. We found a negative effect of work-related identity discrepancy on employee innovation behavior through workplace anxiety. To be specific, our results showed that work-related identity discrepancy has a positive effect on workplace anxiety, which in turn negatively impacts employee innovation behavior. Our findings are consistent with those of previous research, such as Byron and Khazanchi [28] and Eysenck and colleagues [29]. Drawing on solid conceptual and empirical bases, these findings expand the idea that workplace anxiety can affect an abundance of other behaviors relevant to organizational science [24]. We believe that this work constitutes a promising further step toward a more comprehensive understanding of the role of anxiety in the workplace.

Meanwhile, our research sheds light on the comprehension of specific conditions under which the negative effect of work-related identity discrepancy can be mitigated. Drawing on self-discrepancy theory, our research results show that the two types of regulatory focus—promotion regulatory focus and prevention regulatory focus—can influence the effect of work-related identity discrepancy on workplace anxiety and, further, its mediation effect on the relationship between work-related identity and employee innovation behavior. By doing so, this finding contributes back to sociopsychology by suggesting that, in order to curb the negative effect of work-related identity discrepancy, it is necessary for managers to take into account the type of employees’ regulatory focus. It also broadens the boundary of research on the consequences of work-related identity discrepancy by considering the moderation effect of individual traits.

Our research findings carry important implications for organization practitioners as well. First of all, we found that employees who had a close relationship with the previous leader underwent more work-related identity discrepancy. Given this finding, the new leader should pay more attention to the insiders of the previous leader and send them a special friendly signal, either formally (e.g., open-door policy) or informally (e.g., lunch together), to comfort them and prevent them from perceiving work-related identity discrepancy. At the same time, the new leader can consider promoting a more transparent, equitable and healthy relationship with employees. It will give employees a greater sense of security and mitigate the effect of work-related identity discrepancy following the old leader’s departure.

Furthermore, organizations should periodically re-publicize their commitment to equal employment opportunities and highlight that all employees should be treated fairly during leadership transition. Meanwhile, considering the mediation effect of workplace anxiety between work-related identity discrepancy and employee innovation behavior, organizations might want to provide employee assistance programs, such as counseling services, so as to minimize employees’ workplace anxiety derived from work-related identity discrepancy.

Last, but not least, given our finding that employees with high promotion regulatory focus or low prevention regulatory focus feel more workplace anxiety and subsequently become less engaged in innovation behavior when experiencing work-related identity discrepancy, the new leader should adopt different methods when leading employees with different regulatory focus. For employees with promotion regulatory focus, the new leader could spend more time with them on career planning and development and help them adjust their attitudes and career goals; for those with prevention regulatory focus, the new leader should take a more hands-on leadership style and help them fulfill their job responsibilities and be more productive. All these measures can help minimize employees’ workplace anxiety and promote their innovation behavior.

## 5. Limitations

Like other research in this area, our study has several limitations as well. Firstly, we adopted a cross-sectional research design and measured all our variables at one time only through self-report, which could have resulted in common method bias. Although we took proactive and post-active measures, such as ensuring participants’ confidentiality and adopting statistical control [46] to curb this issue, and our varimax rotation analysis confirmed that common method variance was not significant in our study, there is still a need to revalidate our research model with data from multiple sources and using a time-series design [51] or longitudinal research design [52,53] so as to measure the relationships among variables more accurately.

Secondly, we did not test other possible explanations for our findings. For example, with regard to our finding that work-related identity discrepancy has a negative effect on innovation behavior, an alternative explanation derived from social exchange theory [54] is that it could have resulted from the perceived reduction of organization support rather than increased workplace anxiety. Although we believe that self-discrepancy theory is the most suitable of all alternative explanations, drawing on our conceptual model and empirical data, future research might want to consider examining the alternative explanations more rigorously with broader measures.

Thirdly, we used a self-reported questionnaire to assess employee innovation behavior. Although this measure is quite popular in innovation and creativity research [10,15,43,55], it is merely a subjective proxy for innovation behavior that actually evaluates employees’ beliefs and attitudes more than their objective innovative actions per se. Future studies might want to consider such actual measures as the number of implemented new ideas proposed by an employee per year and/or the total savings created by implementing new ideas per year. Of course, a third-party evaluation by a supervisor or colleague will be more objective than self-report [55].

Lastly, we collected our data only from one region in one country. However, constructs like work-related identity discrepancy and workplace anxiety might be culturally bound. Thus, future research might want to consider applying our conceptual model and research design to other regions and/or countries so as to revalidate our findings.

## Figures and Tables

**Figure 1 ijerph-17-06121-f001:**
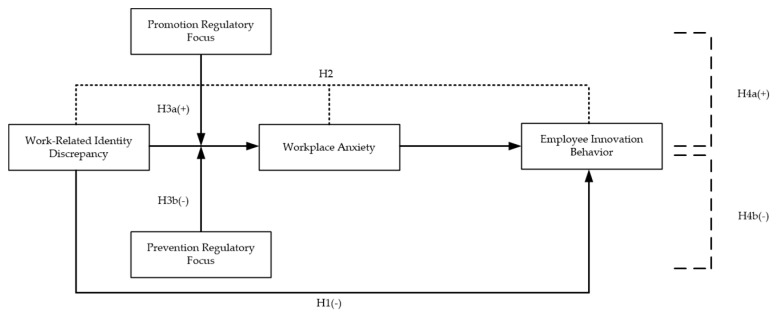
The research conceptual model.

**Figure 2 ijerph-17-06121-f002:**
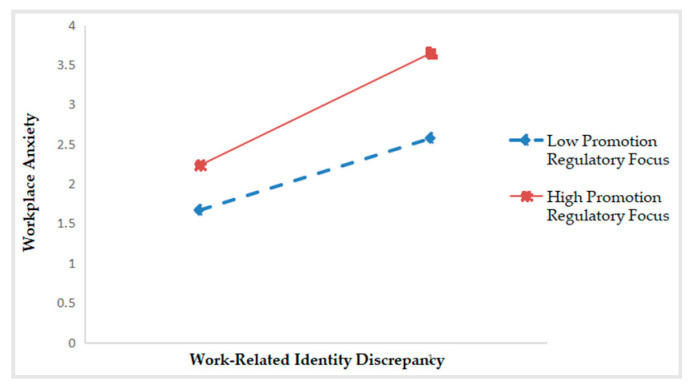
Moderation effect of promotion regulatory focus.

**Figure 3 ijerph-17-06121-f003:**
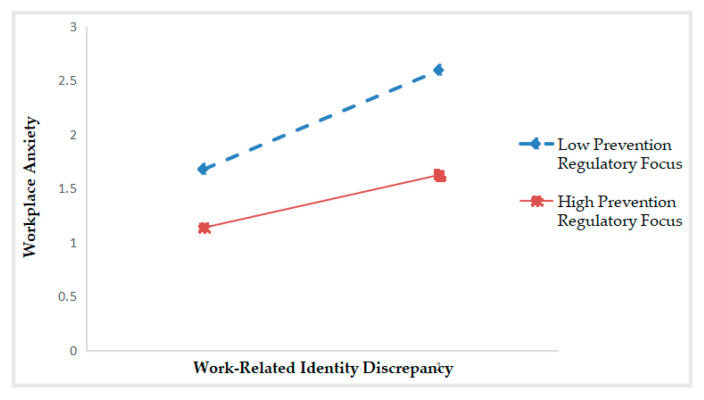
Moderation effect of prevention regulatory focus.

**Table 1 ijerph-17-06121-t001:** Descriptive statistical analysis and correlations (*N* = 563).

Variable	1	2	3	4	5	6	7	8	9
1. Gender									
2. Age	−0.338 **								
3. Education	0.069	0.067							
4. Tenure	−0.222 **	0.588 **	0.091 *						
5. WRID	0.002	0.117 **	0.047	−0.035	(0.865)				
6. WA	0.083 *	0.093 *	0.039	−0.026	0.493 **	(0.928)			
7. P_1_RF	0.024	0.156 **	0.060	0.045	0.279 **	0.296	(0.911)		
8. P_2_RF	−0.002	−0.120 **	−0.016	−0.071	−0.235 **	−0.305	−0.373 **	(0.889)	
9. EIB	−0.038	−0.021	0.022	0.050	−0.501 **	−0.577 **	−0.127 **	0.108 *	(0.891)
M	1.49	33.49	3.20	3.08	2.94	3.16	3.57	3.63	3.56
SD	0.50	8.56	1.09	1.16	1.19	1.09	0.93	0.94	0.91

WRID = Work-Related Identity Discrepancy; WA = Workplace Anxiety; P_1_RF = Promotion Regulatory Focus; P_2_RF = Prevention Regulatory Focus; EIB = Employee Innovation Behavior; Reliabilities (Cronbach’s α) are on the diagonal in parentheses. ** *p* < 0.01, * *p* < 0.05. As for gender, men are coded as 1 and women as 2.

**Table 2 ijerph-17-06121-t002:** Regression results of the moderation test.

Variables	WA				
	M1	M2	M3	M4	M5
**Control**					
Gender	0.123 **	0.093 *	0.097 *	0.105 **	0.090 *
Age	0.203 ***	0.070	0.080	0.091 *	0.089
Education	0.028	0.001	0.008	0.003	0.004
Tenure	−0.121 *	−0.039	−0.051	−0.032	−0.042
**Independent**					
WRID		0.438 ***	0.435 ***	0.416 ***	0.422 ***
**Moderator**					
P_1_RF		0.163 ***		0.164 ***	
P_2_RF			−0.196 ***		−0.194 ***
**Interaction**					
WRID×P_1_RF				0.212 ***	
WRID×P_2_RF					−0.180 ***
R^2^	0.026	0.272	0.292	0.315	0.315
ΔR^2^	0.033 **	0.024 ***	0.036 ***	0.044 ***	0.032 ***

WRID = Work-Related Identity Discrepancy; WA = Workplace Anxiety; P_1_RF = Promotion Regulatory Focus; P_2_RF = Prevention Regulatory Focus; *** *p* < 0.001, ** *p* < 0.01, * *p* < 0.05.

**Table 3 ijerph-17-06121-t003:** Moderated mediation results.

Variable	Level	Conditional Indirect Effect	Upper Level Confidence Interval	Lower Level Confidence Interval
WRID(X)→WA(M)→EIB(Y)	High P_1_RF	−0.052	−0.094	−0.014
	Low P_1_RF	−0.198	−0.248	−0.151
	High P_2_RF	−0.209	−0.261	−0.162
	Low P_2_RF	−0.077	−0.121	−0.038

WRID = Work-Related Identity Discrepancy; WA = Workplace Anxiety; P_1_RF = Promotion Regulatory Focus; P_2_RF = Prevention Regulatory Focus; EIB = Employee Innovation Behavior.

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
