# Peer review of "The Moderated-Mediation Effect of Workplace Anxiety and Regulatory Focus in the Relationship between Work-Related Identity Discrepancy and Employee Innovation"

_ijerph, 2020, doi:10.3390/ijerph17176121_

Round 1
Reviewer 1 Report
Review for International Journal of Environmental Research and Public Health
Comments for Authors
Manuscript ID: ijerph-899430 (revised paper)
Title: The Moderated-Mediation Effect of Workplace Anxiety and Regulatory Focus in the Relationship between Work-related Identity Discrepancy and Employee Innovation
The authors have revised the paper making it shorter and more focused and have responded to the reviewers’ queries.
Two problems remain:
- The authors intended to examine the ‘behaviour’ of the employees in the study. They measured ‘employee innovation behaviour’ using a questionnaire. Strictly, they were not measuring employee behaviour but the employees beliefs about their behaviour. They should mention this in the discussion section.
- In my previous review I pointed out the limitations of this paper suggesting it really a pilot study. As it is a cross-sectional survey, the study is speculative and further studies are needed to rigorously test the authors hypotheses. The authors should add reference to this and what they think might be done in future studies by using more satisfactory methods and measures.
Reviewer 2 Report
The study focusses on employee’s work-related identity impact on behaviours. The study collected data from 563 participants experiencing leadership change and predicts innovation behaviour through anxiety.
The authors have focused positively on the reviewers’ comments and drastically improved the document. It reads easier and functions as an information tools, as journal articles are suppose to do, but has not lost the essence of academia.
The authors drastically shortened the article and improved language and technical challenges. I am satisfied with the authors explanation of the hypothesis and admit that it could have been a misunderstanding (misreading) from my side. I learned from this and would like to thank the authors for their explanation and not just changing the article to suit comments.
The methodology is sound and explanation and results improved. The results and discussion contribute to the body of knowledge.
I wish the authors well in their next endeavor.
Reviewer 3 Report
Dear authors, your article is interesting but I need you to answer some questions:
INTRODUCTION
- Page 1, line 22: It is not written “VUCA (volatility, uncertainty, complexity and ambiguity”, it is written like this “Volatility, Uncertainty, Complexity and Ambiguity (VUCA)”.
MATERIALS AND METHODS
- The authors consulted the ethics committee. You must say the code of the registered protocol.
RESULTS
- The results of the tables are wrong. The authors say that the p values are significant, but are not <0.05 (Table 1 and Table 2). Authors should review the data in the tables.
REFERENCES
Many bibliographies are obsolete and some citations are incomplete. The bibliographic citations used are more than 5 years old (58,8%). The authors must update and arrange
Round 2
Reviewer 1 Report
Review for International Journal of Environmental Research and Public Health
Comments for Authors
Manuscript ID: ijerph-899430 (revised paper2)
Title: The Moderated-Mediation Effect of Workplace Anxiety and Regulatory Focus in the Relationship between Work-related Identity Discrepancy and Employee Innovation
The authors have satisfactorily addressed my remarks.
Author Response
Thank you.

Reviewer 3 Report
Dear authors,
They are right, I did not see some changes that you made.
The way to present the tables is somewhat confusing. I suggest that you try to simplify them.
I do not agree with the opinion of the other reviewer. That percentage of old references is very high. Thank you for respecting my criteria.
Congratulations on your work.
Best regards
Author Response
Thank you.

This manuscript is a resubmission of an earlier submission. The following is a list of the peer review reports and author responses from that submission.
Round 1
Reviewer 1 Report
Review for International Journal of Environmental Research and Public Health
Comments for Authors
Manuscript ID: ijerph-862898
Title: The Moderated-Mediation Effect of Workplace Anxiety and Regulatory Focus in the Relationship between Work-related Identity Discrepancy and Employee Innovation
This paper reports on a cross-sectional survey the data from which is used to test four hypotheses examining the relationship between four variables: workplace anxiety, regulatory focus, work-related identity discrepancy and employee innovation.
The topic of this paper is potentially interesting, but there are some problems with the design of the study:
- The authors have used a cross-sectional survey to examine a complex relationship between four variables. A before and after study or case-control study would have been more appropriate.
- The authors used a convenience sample from one site – a systematic sample would have been more appropriate using different work sites.
This paper would benefit from considerable shortening of length and a clearer focus on the fact that this is really a pilot study.
Reviewer 2 Report
The study investigates identity confusion (?) / discrepancies and/or behaviour or emotional challenges experienced during leadership change in the workplace. The authors posit that employees experience identity discrepancy when managers change and they spend a lot of time impressing the new manager/leader.
The study vaguely addresses an important topic that will provide valuable solutions to workplace productivity. The strongest point of this study is the methodology, which is sound, thorough and clearly set out. The study explains the theoretically implications of the study and is set out well. Most references are current and a fair mix of old and newer sources were used.
Major points needing clarification, etc.
- The introduction makes no reference to what methodology/theory/perspective the research is coming from.
- The document needs to be edited for language and technical consistency.
- The six hypotheses make the study clumsy and overcomplicated. I suggest that the authors condense this to one or two and rather split the article in to two or three other papers.
- Line 210 to 211 refer to “controlled” responses, whilst I am certain this is a translation issue, it raises red flags with regards to the methodology.
- The study needs to address issues of short definitions or descriptions of terms, such as “promotion regulatory focus”, etc. If one is not familiar with the concepts, it makes following the study very difficult and it will make following the results easier.
- The study refers to “behavior” consequences, but this is not unpacked within the introduction, nor any reference in methods or both within Methods or Discussion. It would be nice if there could be a description of behaviors or examples.
Minor points
- Line 25 – 27 is poorly structured and makes no sense.
- Line 39 – “intention” needs to be clarified.
- Line 54 – 56 is vague and difficult to read.
- Line 57 – 59 refers to “different degree of innovation activities” is vague and needs further clarification.
- Line 76 – 79 is again confusing and disorganized.
- Line 82 – 83 makes no sense and I am unsure as to what the authors mean.
Reviewer 3 Report
Dear authors, your article is interesting but I need you to answer some questions:
INTRODUCTION
- Page 1, line 22: It is not written “VUCA (volatility, uncertainty, complexity and ambiguity”, it is written like this “Volatility, Uncertainty, Complexity and Ambiguity (VUCA)”.
- The introduction is very long and should not be to have subsections.
- Sections 1) and 2) must go together. There should be a single "Introduction" section.
MATERIALS AND METHODS
- Authors must specify the research design.
- What was the target population? How was the sample chosen? Authors must specify it.
- The authors must include the response rate of the participants in the study.
- Authors should provide evidence of validity of the scale.
- Have you consulted the ethics committee? Authors must mention and say the reference.
RESULTS
- The results of the tables are wrong. The authors say that the p values are significant, but are not <0.05.
REFERENCES
Many bibliographies are obsolete and some citations are incomplete. The bibliographic citations used are more than 5 years old (58,8%). The authors must update and arrange the bibliography.